# Reduced Placental CD24 in Preterm Preeclampsia Is an Indicator for a Failure of Immune Tolerance

**DOI:** 10.3390/ijms22158045

**Published:** 2021-07-28

**Authors:** Marei Sammar, Monika Siwetz, Hamutal Meiri, Adi Sharabi-Nov, Peter Altevogt, Berthold Huppertz

**Affiliations:** 1Prof. Ephraim Katzir’s Department of Biotechnology Engineering, ORT Braude College, 51 Snunit St, Karmiel 2161002, Israel; 2Division of Cell Biology, Histology and Embryology, Gottfried Schatz Research Center, Medical University of Graz, Neue Stiftingtalstr. 6/II, 8010 Graz, Austria; monika.siwetz@medunigraz.at (M.S.); berthold.huppertz@medunigraz.at (B.H.); 3Hylabs, Rehovot and TeleMarpe, 21 Beit El St., Tel Aviv 6908742, Israel; hamutal62@hotmail.com; 4Ziv Medical Center, Safed, and Tel Hai College, Tel Hai 1220800, Israel; adi_nov@hotmail.com; 5Skin Cancer Unit, DKFZ and Department of Dermatology, Venereology and Allergology, University Medical Center Mannheim, Ruprecht-Karl University of Heidelberg, Theodor-Kutzer-Ufer 1–3, 68167 Mannheim, Germany; p.altevogt@dkfz-heidelberg.de

**Keywords:** CD24, preeclampsia, placenta, cytotrophoblast, syncytiotrophoblast, immunohistochemistry, quantitative polymerase chain reaction (qPCR), immune tolerance

## Abstract

Introduction: CD24 is a mucin-like glycoprotein expressed at the surface of hematopoietic and tumor cells and was recently shown to be expressed in the first trimester placenta. As it was postulated as an immune suppressor, CD24 may contribute to maternal immune tolerance to the growing fetus. Preeclampsia (PE), a major pregnancy complication, is linked to reduced immune tolerance. Here, we explored the expression of CD24 in PE placenta in preterm and term cases. Methods: Placentas were derived from first and early second trimester social terminations (N = 43), and third trimester normal term delivery (N = 67), preterm PE (N = 18), and preterm delivery (PTD) (N = 6). CD24 expression was determined by quantitative polymerase chain reaction (qPCR) and Western blotting. A smaller cohort included 3–5 subjects each of term and early PE, and term and preterm delivery controls analyzed by immunohistochemistry. Results: A higher expression (2.27-fold) of CD24 mRNA was determined in the normal term delivery compared to first and early second trimester cases. The mRNA of preterm PE cases was only higher by 1.31-fold compared to first and early second trimester, while in the age-matched PTD group had a fold increase of 5.72, four times higher compared to preterm PE. The delta cycle threshold (ΔCt) of CD24 mRNA expression in the preterm PE group was inversely correlated with gestational age (r = 0.737) and fetal size (r = 0.623), while correlation of any other group with these parameters was negligible. Western blot analysis revealed that the presence of CD24 protein in placental lysate of preterm PE was significantly reduced compared to term delivery controls (*p* = 0.026). In immunohistochemistry, there was a reduction of CD24 staining in villous trophoblast in preterm PE cases compared to gestational age-matched PTD cases (*p* = 0.042). Staining of PE cases at term was approximately twice higher compared to preterm PE cases (*p* = 0.025) but not different from normal term delivery controls. Conclusion: While higher CD24 mRNA expression levels were determined for normal term delivery compared to earlier pregnancy stages, this expression level was found to be lower in preterm PE cases, and could be said to be linked to reduced immune tolerance in preeclampsia.

## 1. Introduction

The presence of foreign tissues in a host leads to a strong immune rejection directed to destroy the allo-antigen-expressing tissue. Such a response is not developed in normal pregnancy, although half of the fetal genes come from the father and hence are foreign to the mother. The maternal tolerance to the semi-allogeneic fetus enables a normal fetal development that is achieved due to multiple mechanisms [1,2]. This unique example of immune system suppression to hinder a destructive allo-immune response is supported by several placenta-specific genes including proteins such as the signal transducing CD24 protein [3] and placental protein 13 [4]. Many studies have drawn correlations between immune tolerance during pregnancy and tolerance to malignant tumor growth with respect to proliferation, invasion, and immune modulation [5]. Accordingly, many cell pathways used by tumors were evaluated for their function in normal and complicated placental development and vice versa [3].

The CD24 protein is made of a small core of 27 amino acids, which is attached to the membrane via a glycosylphosphatidylinositol (GP-I) anchor [6,7]. It has many potential glycosylation sites for N- and O-linked carbohydrates, rendering the molecule structurally similar to mucins [8]. P-selectin was identified to bind to CD24 under physiological conditions supporting the adhesion of leukocytes to endothelial cells and platelets [8,9,10]. Siglec-10 was found to serve as an additional receptor for CD24 in the immune system [11]. The Siglec10–CD24 binding axis was demonstrated to be an important immune checkpoint for the induction of immune tolerance in mouse autoimmune models [12]. Importantly, recombinant CD24 fusion protein bound to the fragment crystallizable (Fc) region of the immunoglobulin (CD24-Fc) was recently demonstrated to be a powerful drug for blocking overshooting immune reactions (cytokine storm) in SARS-2-COVID-19 infections [13].

In a previous study, we suggested that CD24 may provide similar immune-suppressive features to the developing placenta during pregnancy. We showed that CD24 is expressed in glandular epithelial cells of uterine glands and in other decidual cells [14]. The expression was found to be in close vicinity to invasive extravillous trophoblasts [14]. Additionally, as in tumor cells, first trimester CD24 was found to be co-expressed with Siglec 10 [15,16].

Preeclampsia (PE) is one of the major pregnancy complications. Annually, this life-threatening complication affects 10 million pregnant women globally. It is characterized by hypertension and proteinuria and/or organ failure, requiring earlier delivery to save women’s lives from seizures and stroke (eclampsia). If developed early, inducing premature delivery is often required to save the mother’s life. However, premature delivery may in turn be accompanied by fetal loss and major newborn disabilities [17,18].

The origin of PE, representing a multifactorial disorder, is unknown, but several studies have suggested that it is tightly linked to the loss of maternal immune tolerance to the growing fetus [4,19,20]. When the rejection develops early in pregnancy, it often leads to miscarriage and pregnancy loss [4,21,22]. Overall, immune rejection is implicated as an important process, leading to the clinical features of preeclampsia [19,20,23]. About 70% of all PE cases show clinical symptoms around term (late PE), while about one-third of the cases develop before term (preterm PE, before 37 weeks gestation), of which a minor fraction of cases develops symptoms and is delivered before 34 weeks (early PE) [17,18]. The latter are typically very severe cases and often accompanied by fetal growth restriction (FGR) requiring admission to intensive care units (NICU) due to lower birthweight and many other complications [24,25,26].

In this study, we explored the expression and distribution of CD24 in placentas from the first and early second trimesters and compared them to cases of the third trimester including cases of normal term delivery controls, preterm delivery (PTD), and preterm (≤37 weeks) PE. A smaller cohort was used to compare the latter third trimester cases also to PE cases at term (≥37 weeks).

## 2. Results

### 2.1. The Cohort for Testing CD24 mRNA Expression

The first trimester and early second trimester placentas were derived from social termination of pregnancy at a mean gestational age of 8.6 weeks (Table 1). The control group of third trimester placentas collected at delivery had a mean gestational week at delivery of 38.3 weeks compared to 30.1 weeks in the preterm PE group and 32.8 weeks in the age-matched PTD group. Blood pressure, mean arterial blood pressure, and proteinuria were significantly higher in the preterm PE group compared to any of the other groups. Birthweight was significantly lower for both the preterm PE and PTD groups compared to term controls. However, in the preterm PE group birthweights were at the lower 10th centile, while all babies of the PTD group were >45% centile according to the gestational age (Table 1).

### 2.2. Expression of CD24 mRNA in Placental Tissues throughout Pregnancy

Real-time qPCR analysis revealed a 2.27-fold increase of CD24 in normal term delivery cases compared to 5.72-fold increase in the PTD group and 1.31-fold in preterm PE, all compared to the first and early second trimester group (Table 2, Figure 1). There was a large diversity among individuals in any group. The results are thus best described as a tendency towards increased expression when the first and early second trimester expression level is evaluated versus normal term delivery, a much larger tendency for increase when compared to PTD, and a much lower tendency towards decreased expression in preterm PE, especially compared to the age-matched PTD cases. 

### 2.3. Changes in CD24 mRNA Expression in Preterm Preeclampsia

For the preterm PE group, sharp inverse relationships were identified when the relative placental CD24 mRNA expression (ΔCT) was plotted against the fetal growth centile (Figure 2 left), with a correlation coefficient of R^2^ = 0.388. The ratio of ΔCT standardized to fetal growth (ΔCT/growth centile) plotted against gestational age yielded an R^2^ = 0.545 (Figure 2 right). These results indicated a direct correlation between CD24 mRNA level, fetal growth, and gestational age. There was no correlation for any of the other groups (R^2^ < 0.1, Figure 2 left and right). Accordingly, only in cases of preterm PE did the CD24 mRNA expression increase with fetal weight and gestational age, while in all the other groups, including the age-matched PTD cases, no correlation between CD24 expression and fetal growth or gestational age could be identified. The variance of CD24 mRNA expression in the four groups is depicted in Figure 2.

### 2.4. The Cohort for Testing Placental CD24 Protein Localization

The comparison of CD24 immunostaining in term and early PE became available through a smaller cohort collected at the Medical University of Graz, Austria (Table 3). In this cohort, the early PE cases and the late PE cases had higher blood pressure and proteinuria as anticipated from the definition of this complication (Table 3) compared to their matched preterm and term delivery groups. The early and late PE cases had similar disease severity in terms of elevated blood pressure and proteinuria, and the difference in severity was due to earlier delivery and lower birthweight (Table 3).

### 2.5. Pattern of Immunostaining with CD24-Specific Antibodies

Our previous study [14] has shown that in the first trimester, there is clear immunolabeling for CD24 in the villous and extravillous cytotrophoblasts with a lower staining intensity in villous stroma cells. Interestingly, the protein was not found to be localized in the first trimester syncytiotrophoblast. In this study, cases from preterm delivery as well as normal term and late PE cases showed a varying degree of staining in the syncytiotrophoblast with only minor staining of villous cytotrophoblasts. By contrast, in cases of the early PE group, there was a strong staining of the villous cytotrophoblasts with only minor staining of the syncytiotrophoblast (Figure 3A–D).

Looking at the images of this study in detail, it appears as if in the PTD (Figure 3B1) and normal term delivery group (Figure 3D1) there was a moderate staining of the syncytiotrophoblast (blue arrows) and light staining of villous blood vessels (red arrows, Figure 3B2,D2), with very little staining of the villous cytotrophoblast (green arrow, Figure 3B1,D1). In late PE, staining of the syncytiotrophoblast showed very strong impregnation (blue arrow, Figure 3C1), while blood vessels staining remained very weak (Figure 3C2). By contrast, in the early PE cases, as in first trimester placenta, the villous cytotrophoblasts were stained (green arrow, Figure 3A1), while the syncytiotrophoblast remained mostly negative or pale (blue arrow, Figure 3A1).

The strongest staining for CD24 was found in the villous trophoblast layer of cases with late PE (Figure 3C) compared to moderate staining in PTD (Figure 3B) and normal term cases (Figure 3D). CD24 staining in cases from the early PE group was very low (Figure 3A). Villous stroma and blood vessels showed light staining for CD24, and there was no clear staining pattern (Figure 3A–D, red arrows).

### 2.6. Semi-Quantitative Analysis of CD24 Staining

Villous trophoblast. CD24 staining was significantly reduced in early PE cases compared to age-matched preterm delivery cases (*p* = 0.042) (Figure 4A, Table 4a). Staining was mainly found in villous cytotrophoblast but not in the syncytiotrophoblast, which could account for the reduced overall staining intensity in the early PE cases. Staining for CD24 in late PE was nearly two times higher compared to early PE (*p* = 0.025) (Figure 4A, Table 4a). Staining for CD24 in late PE placental tissues was not different from the PTD group or normal term delivery (*p* = 0.143) (Figure 4A, Table 4a,b). The comparison of all unaffected cases (term and preterm delivery combined) to all PE cases (early and late PE cases combined) did not reveal any difference. Thus, it appears that unaffected cases were similar in staining intensity, while in the case of the PE subjects, the lower values in early PE were compensated by the higher values in the late PE group (Figure 4A, Table 4c). Altogether, lower trophoblast staining for the CD24 protein in early PE corresponded to lower CD24 mRNA expression in preterm PE as described in Figure 2 and Table 2. 

Stroma and vessels. We did not identify any significant differences for any comparison between the groups (Figure 4B,C, Table 4).

### 2.7. CD24 Protein Determination in Placental Lysate

We further determined CD24 protein in lysates of placental tissue derived from preterm preeclamptic and term delivery controls by Western blot analysis. These immunoblots revealed diffuse bands of CD24 with molecular weights ranging between 30 and 60 kDa (Figure 5A). The diffuse bands from the total placental lysate reflected the high degree of CD24 glycosylation. Semi-quantitative analysis by densitometry revealed a significantly reduced CD24 protein expression (*p* = 0.026) in placentas derived from preterm PE cases compared to term controls (Figure 5B). 

There was some diversity among the PE cases (Figure 5A), which could not be attributed to any of the following: disease severity (hypertension, proteinuria), maternal or gestational age, or newborn birthweight. Answer: The preterm group was composed of 14 early cases (delivered before 34 weeks) and 4 preterm cases (delivered between 34 weeks and 0 days and 36 weeks and 6 days). All four cases in Figure 5A were early PE cases, and the calculation for Figure 5B was made for the entire group and for the early and preterm subtypes separately. The calculations did not show any differences between the two subtypes.

## 3. Discussion

We have previously reported expression of CD24 in the first trimester placenta [14]. In the present study, we extended the study by showing firstly that the level of CD24 increased from the first and early second trimesters compared to term delivery as assessed by qRT-PCR analysis. Reduced CD24 mRNA expression in the placenta from PE cases has already been reported [27]. Secondly, we compared CD24 mRNA expression of term delivery to preterm PE and the aged-matched PTD group and found that it is lower in preterm PE compared to normal term delivery and especially compared to age-matched PTD. Thirdly, immunohistochemistry revealed reduced expression of CD24 protein in early PE, corresponding to reduced mRNA and protein expression in placental lysates from preterm PE cases. Altogether, our different methods of assessment indicate reduced CD24 expression in cases of early and preterm PE. The fourth finding was the increased level of CD24 protein expression in term PE cases compared to term controls, providing additional evidence that the two types of PE (early and preterm versus term), although similar in many clinical features, may be associated with different underlying pathways as was already suggested by others [18,28,29].

CD24 labeling of the villous syncytiotrophoblast was noted in normal term control specimens. A similar staining pattern was previously identified in a study by McDonald et al. [30]. The presence of CD24 in the syncytiotrophoblast in normal term delivery may be associated with placental maturation. Staining of other regions of the placenta at term delivery was very light. In comparison, we have already shown that during the first trimester, and already in gestational week eight [14], CD24 showed high expression in villous and extravillous cytotrophoblasts. One may ask why there is such a strong labeling at this period in the villous cytotrophoblasts, and especially in endometrial glands and extravillous trophoblasts in the placental bed. It appears that such labeling at that period may indicate the frontier of immune tolerance linked to escape rejection of the invading trophoblasts during the first trimester [4,28]. In comparison, the labeling of the syncytiotrophoblast in the third trimester and the very high level in late PE may indicate the frontier in immune tolerance to be related to this component of the placenta, as was demonstrated with another immune tolerance molecule in this period [31].

Understanding the pathways underlying immune tolerance in pregnancy has important implications as it may allow for the development of new ways to prevent pregnancy complications such as PE and also FGR, as was shown by Guleria et al. [32]. A variety of molecular and cellular processes were described, aiming to explain how immune tolerance is established [33,34]. In previous works, using placental protein 13 (PP13) as a marker [4], we have suggested that the development of early/preterm PE may be linked to a reduced immune tolerance at the feto–maternal interphase. In this respect, we proposed that placental proteins such as PP13 may render the mother immune-tolerant to the invading trophoblasts by inducing apoptosis of the decidual leukocytes, particularly T-cells [35,36]. We suggested that reduced first trimester levels of PP13, either due to primary structure mutation, promotor polymorphism, or reduced mRNA expression, could well be associated with an increased risk to develop PE [37].

This may also explain the difference in CD24 localization between the first trimester and term delivery. Accordingly, during the period of placentation (first trimester), the immune suppression impact of CD24 may be required to escape rejection of the invading trophoblasts [4,29,30]. In comparison, near delivery CD24 labeling of the syncytiotrophoblast might indicate requirements to protect the supply of nutrients and oxygen from the mother to the placenta and the fetus.

In the present study, we evaluated CD24 expression by comparing the protein labeling by immunohistochemistry in early, preterm, and term PE cases. We assumed that altered levels of CD24 could affect immune tolerance via the Siglec10-CD24 binding axis [15,16]. Indeed, we observed that in early/preterm cases of PE, there is a tendency of reduced mRNA and protein expression of CD24, as well as a significantly lower intensity of placental labeling, especially in the villous syncytiotrophoblast of the placenta. This could be an indication for a process of reduced immune tolerance in the context of early/preterm PE. Interestingly, we found that CD24 labeling of the syncytiotrophoblast is increased in PE that are developed around term.

Today, there is no universal agreement as to the differences underlying the pathways leading to the development of PE subtypes, whether term, preterm, or early. The definition is mainly related to the time the clinical situation requires delivery, as the time of onset is usually not clearly defined. The pathological borderline or marker differences between early cases (before 34 weeks) and preterm cases (between 34 weeks and 0 days and 36 weeks and 6 days) are very blurred. By contrast, the differences between the two subgroups of preterm and term PE are quite clear. The main differences between preterm and term PE can be found (1) with regards to blood vessel remodeling as estimating by measuring the blood flow through the uterine arteries and detected by the increased Doppler pulsatility Index (UTPI) and (2) with regards to a larger imbalance between soluble FMF-like tyrosine kinase-1 (sFlt-1) and placental growth factor (PlGF). In addition, aspirin prevention of PE has a relative success (62%) in preterm but not term PE cases [18,28,29,38].

The difference in distribution of CD24 labeling in term PE versus early PE may be linked to an underlying process related to coagulation. In fact, in certain cases of PE around term a strong process of coagulation is identified in the placental villi. Increased P-selectin and CD24 in these cases may be linked to enhanced activation of platelets and endothelial cells linked to blood vessel activation, all of which are part of the process of hypercoagulation that often occurs in term PE cases. This may create another resemblance between PE and tumors [8,9,10].

In human tumors, CD24 has been introduced as a diagnostic and prognostic marker [39,40]. Strong CD24 expression was linked to a rapid tumor progression in epithelial ovarian cancer, breast cancer, non-small cell lung carcinoma, prostate cancer, and pancreatic cancer [40,41]. Additionally, CD24 expression analysis allowed for the prediction of metastasis in malignant melanoma [41,42].

The role of CD24 in inducing immune tolerance was initially investigated in the immune system but later also in tumors [40,41,42,43]. It is interesting to note that the CD24–Siglec-10 binding axis has adverse effects in human cancer [15,16]. Barkal et al. [44] demonstrated that CD24 is a potent anti-phagocytic “don’t eat me” signal [43] that is capable of directly protecting cancer cells from attack by Siglec-10-expressing macrophages. Monoclonal antibody blockade of CD24–Siglec-10 signaling robustly enhances clearance of CD24+ tumors, which indicates promise for CD24 blockade in immunotherapy. This is the underlying rational for new drug developments such as EXO-CD24 (N. Arber, Israel) or Merck’s CD24-Fc to dampen overshooting immune reactions as observed during the COVID-19-related cytokine storm.

We have focused on villous tissues (the “fetal part” of the placenta) as this tissue represents the largest surface of fetal tissues in direct contact to maternal blood and all the circulating immune cells. We anticipate that in the placenta, there is a potential blockade of CD24, resulting in blocking macrophage attack of placental villi and increased immune tolerance, as is the case in tumors [43,44]. It is tempting to speculate that during the normal course of pregnancy, CD24 blocks macrophages and thereby supports the growth of the placenta. Therefore, CD24 reduction in early PE may be linked to increased rejection of the placenta, and its reduced expression may be associated with early cases of PE and could be relevant to cases also complicated by fetal growth restriction.

Today, only one additional study explored CD24 in the placenta of PE cases [27]. Our study is thus opening a whole new line of research in evaluating the role of this protein the placenta. We hope our manuscript will stimulate other scientists to develop CD24 knockout/knockdown models to further illuminate the role of this protein in the placenta in general and in particularly in immune tolerance during pregnancy.

The main limitations of our present study are as follows: CD24 was not determined at fixed time points, but rather when the patients were admitted to the hospital; however, this reflects clinical reality. A second limitation is that the design of the study was such that we did cross-sectional and not repeated measurements during pregnancy, which have been shown to improve accuracy. A third limitation is that although this is a first study to show reduced CD24 mRNA and protein expression as well as immune staining in preterm PE, the sample size was limited, and additional studies are warranted to add power and strength to our findings. The fourth limitation is that the comparison between term and preterm PE looks promising but requires more diversified analysis of the two PE subtypes. Finally, we did not evaluate the release of this protein from the placenta into the maternal circulation for future use of predictive estimates with simple blood tests. Huppertz et al. [45] have demonstrated the importance of obtaining results of a placental protein from diversified origins (placenta, serum, amniotic fluid) to reach a comprehensive understanding of the actual origin of the measured protein and further verify its role in the placental pathology in PE [45,46,47].

In conclusion, this is the first study to systematically evaluate placental CD24 expression in the first trimester compared to preterm and term delivery (early and late third trimester). Our study found an increased expression level during pregnancy, and the shift from staining of placental bed towards intensified villous staining. Our study showed decreased CD24 mRNA and protein expression and changes in protein localization in placentas from early and preterm PE cases corresponding to reduced immune tolerance. Higher CD24 staining in PE at term might be linked to hypercoagulation and may indicate a potential difference in the underlying pathways. The role of altered levels of CD24 in pregnancy should be further evaluated in the context of immune tolerance as a marker and potentially also as a new therapy target. 

## 4. Materials and Methods

### 4.1. Antibodies

The monoclonal antibody (mAb) clone SWA11 was used for the detection of CD24 [48,49,50]. This mAb is specific for CD24 and reacts with the leucine–alanine–proline (LAP) motif in the protein core, as shown by peptide inhibition studies [49]. In addition, SWA11 exhibits specific binding to CD24-transfected cells but not to vector control [50]. Anti β-actin (clone 4) was purchased from MP Biomedical (Santa Ana, CA, USA).

### 4.2. Specimens

Two cohorts were used for this study. The first was a cohort of patients who donated their placentas according to the ethics approval of the Institutional Review Committee (IRB) of Bnai Zion Medical Center, Haifa, Israel (#BZ-06-021-972). This cohort included placentas from social terminations in the first and early second trimesters of pregnancy (N = 43), placentas from normal term delivery control (N = 67), and preterm PE (N = 18) and preterm delivery (PTD) (N = 6) patients. The preterm PE patients of this groups can be subdivided into 14 early cases (delivered before 34 weeks) and 4 preterm cases (delivered between 34 weeks and 0 days and 36 weeks and 6 days). However, there was no significant difference between the two subgroups and none of the results pointed to the need to subdivide them for improving group description or clarity of the results.

A small complementary cohort was collected at the Department of Obstetrics and Gynecology of the Medical University of Graz, Austria, after obtaining the approval of the ethics committee of the Medical University of Graz, Austria (# 24-112 ex 11/12). This cohort included placentas from normal term delivery controls (N = 5), late PE (N = 3), early PE (N = 3), and preterm delivery (PTD) (N = 3) patients. All women signed a written informed consent form before providing their specimens. The Israelian cohort was used for PCR and Western blotting, whereas the Austrian cohort was used for immunohistochemistry.

### 4.3. Definition of Pregnancy Complications

First and early second trimester social terminations were approved by the Hospital Pregnancy Termination Committee according to the National Law in Israel. Gestational age was determined according to last menstrual period and verified by measurements of crown–rump length (CRL) [51]. Normal term delivery control placentas were obtained from women delivering a baby at the 75–100 centile at gestational age of 37–41 weeks.

Preeclampsia (PE): In this paper, we used the updated criteria for the definition of preeclampsia as published in June 2020 by the American College of Obstetrics and Gynecology (ACOG) and of the International Society for the Study of Hypertension Disorders of Pregnancy (ISSHP) [18,52]. PE was defined as systolic blood pressure of 140 mm Hg or more or diastolic blood pressure of 90 mm Hg or more on two occasions at least 4 h apart after 20 weeks of gestation in a woman with a previously normal blood pressure. New onset proteinuria was defined as 300 mg or more per 24 h urine collection (or this amount extrapolated from a timed collection) or protein/creatinine ratio of 0.3 mg/dl or more or dipstick reading of 2+ (used only if other quantitative methods were not available) [52,53]. In the absence of proteinuria, new-onset hypertension with the new onset of any of the following were taken into account: (1) thrombocytopenia (platelet count less than 100 × 10^9^/L), (2) renal insufficiency (serum creatinine concentrations greater than 1.1 mg/dl or doubling of the serum creatinine concentration in the absence of other renal disease), and/or (3) impaired liver function (elevated blood concentrations of liver transaminases to twice the normal concentration) [54,55,56]. Other symptoms included pulmonary edema, new-onset headache unresponsive to medication, and those not accounted for by alternative diagnoses or visual symptoms [18,52]. Given the new ACOG and ISSHP definition of PE published after the study was completed, we reviewed the database on a patient-by-patient basis to verify that patients included in the PE group according to our hospital clinical guidelines comply with these new ACOG and ISSHP definitions [18,52]. 

Small for gestational age (SGA) and fetal growth restriction (FGR) were defined according to the criteria of the international society for ultrasound in obstetric gynecology (ISUOG) [24] as the estimated fetal weight < 10th percentile and according to abdominal circumference < 10th percentile (SGA) [24,25,26]. When SGA was combined with oligohydramnion (AFI < 5 cm) and/or high pulsatility index of the blood flow through the umbilical cord and middle cerebral arteries and the ductus venosus at the highest (>90th percentile) level, it was defined as FGR [24]. Estimation of fetal weight was made with the use of head, body, and femur measurements [57], as adjusted to the Israelian population [58]. 

Preterm delivery (PTD) was defined as delivery before 37 weeks [59,60] not related to FGR and PE, or placental abruption, contaminated amniotic fluid, or preterm premature rupture of membranes (PPROM).

### 4.4. Immunohistochemistry

Placental samples were formalin-fixed and paraffin-embedded according to standard procedures. Sections (5 µm) were de-paraffinized in xylene and rehydrated in a graded series of alcohol. Epitope retrieval was performed for 7 min at 120 °C using a de-cloaking chamber (Biocare Medical, Pacheco, CA, USA). The procedure was performed with antigen retrieval solution pH 9 (Leica Biosystems, Newcastle, UK).

Sections were immunostained using the UltraVision Detection System HRP Polymer (LabVision, Thermo Fischer Scientific, Kalamazoo, MI, USA) according to the manufacturer’s instruction. In brief, slides were incubated with hydrogen peroxide block to quench endogenous peroxidase. After washing with TBS/T (Tris-buffered saline (pH 7.4) containing 0.05% Tween20; Merck, Darmstadt, Germany), non-specific background was blocked by incubation with UltraVision Protein Block. The anti-CD24 antibody, clone SWA 11, was diluted 1:750 in Antibody Diluent (Dako, Carpinteria, CA, USA) and incubated for 45 min at RT.

After washing with TBS/T, Primary Antibody Enhancer, and after another washing step, UltraVision HRP polymer was applied for 20 min. The polymer complex was visualized by incubating the slides with 3-amino-9-ethylcarbazole (AEC) Chromogen Single Solution (Lab Vision) for 10 min. Sections were counterstained with hemalaun and mounted with Kaiser’s glycerin gelatin (Merck). Controls were performed by using a mouse IgG negative control antibody (NeoMarkers, Thermo Scientific, Waltham, MA, USA) and revealed no staining.

### 4.5. Semi-Quantitative Analysis of Immunohistochemical Staining

For each case, images of a placental tissue section, stained for CD24, were taken in a stereology workstation using a Leica DM 6000B Microscope with an Olympus DP72 Camera and VIS Visiopharm software (Visiopharm A/S, Hoersholm, Denmark) by applying systematic uniform random sampling. For each slide, 10 images were systematically and randomly selected, representing the whole section. Each of these 10 images was analyzed as follows: A dot grid with 16 × 12 dots was placed on each image using the newCAST software (Visiopharm A/S). Each of these dots was assigned to the structure it was lying on top, defined as intervillous space, villous trophoblast, placental blood vessel, or villous stroma. Additionally, the staining intensity was categorized into strong (++), weak (+), or negative (-). Each dot was assigned to the respective combination of categories (tissue type/staining intensity). The sum values of all positive and negative dots of the 10 sections per block was arithmetically calculated and the percentiles of the positive staining were calculated.

### 4.6. Western Blot

Placental tissue processing was performed as described by Sammar et al. [31]. Briefly, placental tissue was homogenized and solubilized in RIPA lysis buffer (1% NP-40, 0.5% sodium deoxycholate, 0.1% SDS, 20 mM Tris/HCl (pH 8.0), 150 mM NaCl) containing a complete set of protease inhibitors (Hoffman la Roche, Switzerland) for 30 min at 2–8 °C. Insoluble material was removed by centrifugation and protein concentration was determined by BCA reagent (Pierce, Rockford, IL, USA). Placental lysate samples were aliquoted and stored at −70 °C until use.

For Western blot analysis, 50 µg of total protein lysates was separated on 12.5% SDS-PAGE and electro-transferred to a nitrocellulose membrane. After blocking free binding sites with 5% non-fat milk in 20 mM Tris/HCl buffer at pH 8.0 supplemented with 150 mM NaCl, membranes were probed with the anti-CD24 mAb SWA11 and anti-β-actin antibodies (0.1 µg/mL) overnight at 4 °C. Bound immune complexes were detected by horseradish peroxidase-conjugated rabbit anti-mouse IgG and developed by ECL detection kit (Biological Industries, Beit Haemek, Israel). β-Actin was used as reference housekeeping protein for equal protein loading and densitometry analysis. Signals were developed by chemiluminescence and were captured by an imager (Bio-Rad, Hercules, CA, USA).

### 4.7. Placental CD24 mRNA Amplification by PCR

Total RNA was isolated from frozen blocks of placental tissues and was reverse transcribed as described before [57]. cDNA synthesis was performed using superscript cDNA synthesis kit (Invitrogen, Carlsbad, CA, USA). Expression of CD24 gene was quantified by TaqMan RT-PCR utilizing the Applied Biosystem StepOne Plus cycler (Applied Biosystems, Austin, TX, USA) and TaqMan Gene Expression Assay with primer and probe sets (Applied Biosystems) for CD24 (Hs02379687_s1). Hypoxanthine-guanine phosphor-ribosyl-transferase (HGPRT) (Hs99999909_m1) was used as housekeeping gene. The relative amount of CD24 was calculated by employing the comparative Ct method (2−ΔΔCt).

### 4.8. Statistics

We used the statistical SPSS version 24 (SPSS Inc., Chicago, IL, USA) for analysis. The differences across groups were calculated by Kruskal–Wallis non-parametric tests (P). Mann–Whitney non-parametric tests were used to compare two groups. Statistical significance was defined when *p* < 0.05. Box-plot graphs provide a visualization of value distribution across quartiles.

The formula for the linear correlation coefficient was calculated according to
$$r_{xy}=\frac{n\sum_{i=1}^{n}x_{i}y_{i}-\sum_{i=1}^{n}x_{i}\sum_{i=1}^{n}y_{i}}{\sqrt{n\sum_{i=1}^{n}x_{i}^{2}-{\left(\sum_{i=1}^{n}x_{i}\right)}^{2}}\sqrt{n\sum_{i=1}^{n}y_{i}^{2}-{\left(\sum_{i=1}^{n}y_{i}\right)}^{2}}}$$
where *n* is the number of specimens analyzed, Σ*x* = total of the growth centile or gestational week (first variable value), Σ*y* = total of ΔCT or ΔCT/growth centile (the second variable value), Σ*xy* = sum of the product of first and second values, Σ*x*^2^ = sum of the squares of the first value, and Σ*y*^2^ = sum of the squares of the second value.

## Figures and Tables

**Figure 1 ijms-22-08045-f001:**
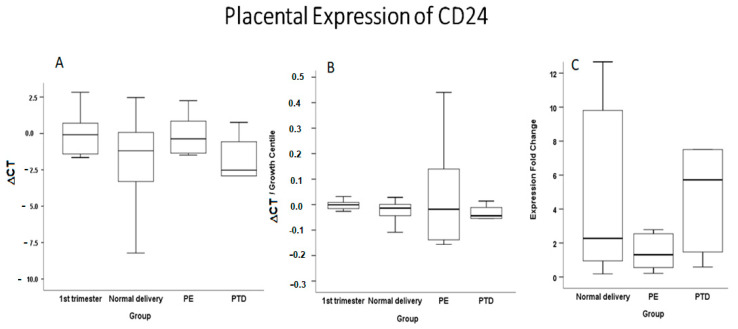
Box plot of placental mRNA expression of CD24. (**A**) ΔCT, (**B**) ΔCT versus fetal growth, (**C**) expression fold (2^−Median^^ΔΔCT^*)* change.

**Figure 2 ijms-22-08045-f002:**
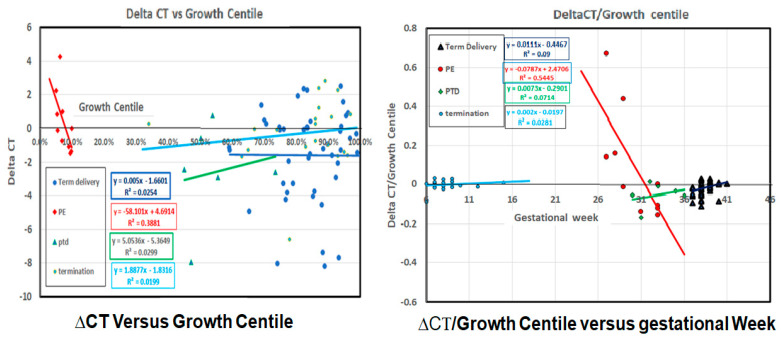
CD24 mRNA expression and fetal growth characteristics. **Left**-ΔCT versus growth centile. The mRNA expression assessed via ΔCT is presented for the various clinical outcome groups, indicating high correlation only for the case of the preterm preeclampsia (PE) group. **Right**-ΔCT/growth centile versus gestational week. The ΔCT/growth centile shows a correlation to gestational week at delivery only for the preterm PE cases. PE-preeclampsia, PTD-preterm delivery. Linear regression for each group is shown with regression curve and coefficient listed in the figure.

**Figure 3 ijms-22-08045-f003:**
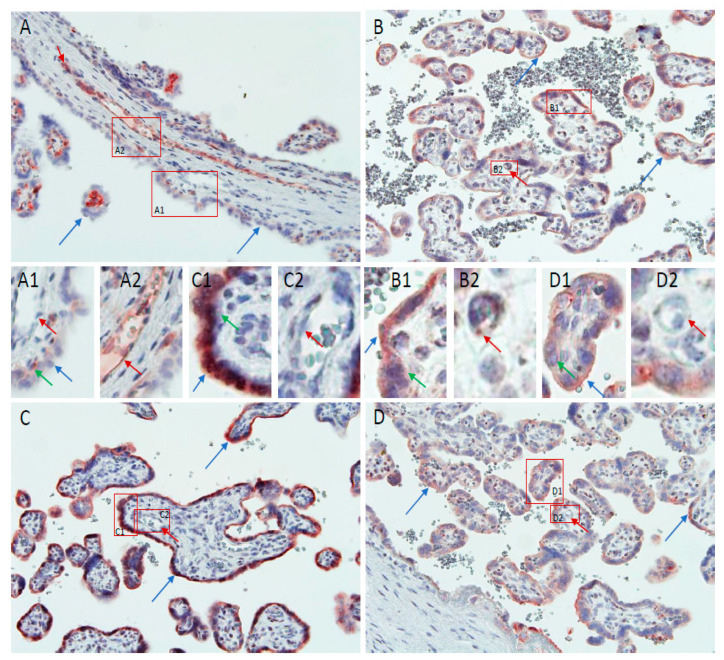
Immunohistochemical staining for CD24 in placental tissues. (**A**) early PE; (**B**) preterm delivery; (**C**) late PE; (**D**) term control. Blue arrows point to villous syncytiotrophoblast, green arrows point to villous cytotrophoblasts, and red arrows to villous blood vessels. Original magnification of A to D × 200. The small images are higher magnifications from the four larger images to better show staining of villous syncytiotrophoblast and cytotrophoblast (**A1**–**D1**), as well as villous blood vessels (**A2**–**D2**).

**Figure 4 ijms-22-08045-f004:**
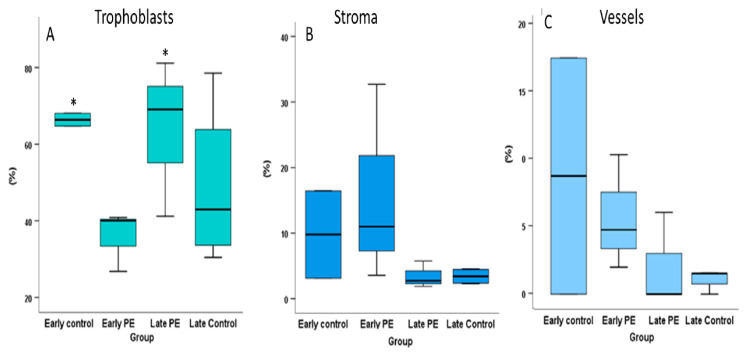
Semi-quantitative analysis of CD24 staining. (**A**) Villous trophoblast, (**B**) stroma, (**C**) vessels. (**A**) Villous trophoblast staining in early preeclampsia cases was lower compared to staining in gestational age-matched preterm delivery cases. In comparison, labeling of term delivery controls was not significantly different to late PE cases. Early PE was significantly lower compared to age-matched PTD cases as well as compared to late PE cases. (**B**,**C**) No significant differences could be detected for villous stroma and placental blood vessels. * *p* < 0.05 by Mann Whitney a-parametric test between the two early and the two term groups.

**Figure 5 ijms-22-08045-f005:**
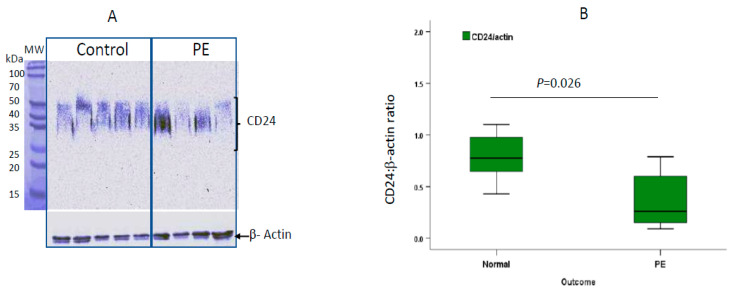
Western blot analysis of placental CD24 protein expression. (**A**) Representative Western blot of CD24 protein expression in the placenta. Placental CD24 was analyzed by Western blotting with a specific anti-CD24 antibody. β-Actin was used as housekeeping protein control for loading of equal amounts of proteins on the gel. (**B**) Semi-quantitative densitometric assessment of CD24 and β-actin bands was performed by Image Gauge software. The intensity of the CD24 bands were normalized to β-actin as a housekeeping protein. The analysis included 8 cases of preterm preeclampsia and 10 term control cases.

**Table 1 ijms-22-08045-t001:** Demographic and pregnancy characterization—Israelian cohort.

Parameter (Mean (Range))	First and Early Second Trimester (*n* = 43)	Normal Term Delivery (*n* = 67)	Preterm PE (<37 Weeks) (*n* = 18)	PTD (<37 Weeks) (*n* = 6)
Gestational age at delivery (weeks)	8.6 (6.0–18.0)	38.3 * (37.0–41.0)	30.1 *^#^ (24.0–36.0)	32.8 *^#^ (30.0–36.0)
Maternal age (years)	27 (19–39)	32 (21–42)	29 (20–47)	30 (20–41)
Ethnicity (Jew, n, %)	40 (93.0%)	56 (84.8%)	13 (72.2%)	5 (83.3%)
Parity	0.9 (0.6–1.2)	1.3 (1.1–1.6)	0.6 ^#^ (0.1–1.0)	1.3 (0.2–2.4)
Growth centile (%)	77 (32–99)	84 (59–108)	7 *^#^ (3–10)	54 *^#^ (45–74)
Urine protein (mg/mL)	NA	NA	3057 ^#^ (0–17,000)	5.0 ^ω^ (0–305)
Systolic BP (mmHg)	100 (89–134)	122 (93–152)	169 *^#^ (134–220)	125 ^ω^ (115–135)
Diastolic BP (mmHg)	69 (60–82)	70 (51–87)	102 *^#^ (81–120)	69 ^ω^ (60–82)
MAP (mmHg)	79 (70–98)	87 (46–109)	125 *^#^ (99–147)	88 ^ω^ (79–100)
Baby’s birthweight (grams)	-	3428 (2500–4290)	1228 ^#^ (470–2520)	1858 ^#ω^ (1135–2645)
Baby’s gender (male, n, %)	-	35 (52%)	7 (41%)	4 (67%)

Values are shown as means and ranges. Mann–Whitney analysis was used to calculate the statistical difference between the groups: significantly different values are marked when *p* < 0.05, as * compared to first trimester, ^#^ as compared to term delivery, and ^ω^ for preterm PE compared to PTD. NA, not available. No gender and weight were determined in social termination.

**Table 2 ijms-22-08045-t002:** Placental CD24 mRNA expression.

Median (95% CI)	First and Early Second Trimester (*n* = 43)	Normal Term Delivery (*n* = 67)	Preterm PE (*n* = 18)	PTD (*n* = 6)	*p*-Value
ΔCT	−0.01 (−1.30–0.71)	−1.19 (−1.79–(−0.09))	−0.37 (−1.36–0.85)	−2.52 (−2.92–(−0.57))	0.151
ΔCT/ growth centile	−0.001 (−0.015–0.008)	−0.014 (−0.021)–(−0.001)	−0.018 (−0.138–0.157)	−0.044 (−0.054)–(−0.011)	0.126
Expression fold change (2^−Median^^ΔΔ^^CT^)	1.00	2.27 (1.05–3.43)	1.31 (0.55–2.54)	5.72 (1.47–7.50)	0.353

ΔCT: expression cycle (CD24)-expression cycle (housekeeping gene). ΔΔCT = ΔCT (X) − median ΔCT (first and early second trimester) (95% CI). *p*-values were calculated by the Kruskal–Wallis a-parametric test between all four groups, which showed no statistical differences. Mann–Whitney comparison between any group pair was insignificant.

**Table 3 ijms-22-08045-t003:** Demographic and pregnancy characterization—Austrian cohort.

Parameter	Early PE (*n* = 3)	PTD (*n* = 3)	Late PE (*n* = 3)	Term Control (*n* = 5)	*p*- Value (#)
Gestational age at delivery (weeks)	32.3 ^b^ (31.0–34.0)	31.3 ^b^ (29.0–34.0)	38.7 ^ab^ (38.0–40.0) ^y^	40.0 ^a^ (39.0–41.0) ^e^	0.013
Proteinuria (g/24 h)	2973 ^a^ (2113–4562) *	0 ^b^ (0–0)	2293 ^ab^ (2293–2293) *	0 ^b^ (0–0)	0.015
Systolic BP (mmHg)	171 (150–196)	NA	176 (148–202) *	109 (90–111)	0.319
Diastolic BP (mmHg)	106 (100–114)	NA	104 (96–116) *	80 (75–85)	0.319
MAP (mmHg)	128 (117–141)	NA	128 (117–145) *	97 (90–9100)	0.319
Mode of delivery (vaginal, n, %)	3 (100%)	3 (100%)	2 (66.7%) *	1 (33.3%)	0.506
Baby’s birthweight (gram)	1267 ^b^ *	2030 ^ab^ (1870–2190)	2939 ^a^ (2398–3630) ^y^	3023 ^a^ (2905–3140) ^e^	0.052
Baby’s gender (male, n, %)	1 (33.3%)	2 (66.7%)	2 (66.7%)	NA	0.836
Growth centile (%)	12.6 (0.9–21.5)	65.2 (51.0–79.3)	51.9 (4.7–90.5)	39.0 (29.9–48.2)	0.218
Placental weight (gram)	350 *	440 ^b^ (350–500)	520 ^ab^ (450–650) *^y^	617 ^a^ (465–720) ^e^	0.041
Placental weight/fetal weight (ratio)	3.7 ^b^ (3.3–4.1) *	5.7 ^a^ (5.3–6.1)	5.0 ^a^ (4.7–5.3) *^y^	4.7 ^ab^ (4.5–4.8)	0.049

Values are shown as means and ranges. * *p* ≤ 0.05 for Mann–Whitney a-parametric test between early PE to PTD or term PE and term delivery. ^y^ *p* ≤ 0.05 for Mann–Whitney a-parametric test between early to late PE. ^e^ *p* ≤ 0.05 for Mann–Whitney a-parametric test between PTD to term delivery. # Kruskal–Wallis a-parametric test between all four groups. In Kruskal–Wallis analysis, “a” is significantly higher from values of other groups, “b” is significantly lower, “ab” is in between “a” and “b”, and “c” is the lowest. NA, not available. *p*-values in bold indicate significant differences.

**Table 4 ijms-22-08045-t004:** Quantitative analysis of CD24 staining.

**(a) Comparison of Trophoblast, Stroma, and Vessel Staining Comparing Each Sub-Pathology to Its Matched Control Group**								
**Median [ranges]**	**Early PE**	**Early Control**	***p* ***	**Late PE**		**Late Control**	***p* ***	***p* ^#^**
Trophoblast, %	35.9 (26.8–40.9)	66.4 (64.7–68.1)	0.042	63.8 (41.2–81.1)		48.7 (30.5–78.6)	0.143	0.175
Stroma, %	15.8 (3.6–32.7)	9.8 (3.1–16.4)	0.282	3.4 (1.9–5.7)		3.4 (2.3–4.6)	0.499	0.300
Vessels, %	5.7 (2.0–10.3)	8.8 (0–17.5)	0.500	2.0 (0–6.1)		1.2 (0–1.6)	0.357	0.369
**(b) Comparison Between the Pathology Groups According to Staining of Trophoblast, Stroma, and Vessels**								
	Early PE	Late PE	*p* *		Early Control	Late Control		*p* *
Trophoblast, %	35.9 (26.8–40.9)	63.8 (41.2–81.1)	0.025		66.4 (64.7–68.1)	48.7 (30.5–78.6)		0.167
Stroma, %	15.8 (3.6–32.7)	3.4 (1.9–5.7)	0.064		9.8 (3.1–16.4)	3.4 (2.3–4.6)		0.177
Vessels, %	5.7 (2.0–10.3)	2.0 (0–6.1)	0.134		8.8 (0–17.5)	1.2 (0–1.6)		0.407
**(c) Comparison of Trophoblast, Stroma, and Vessel Staining According to All PE versus All Unaffected**								
	All PE		All Unaffected			*p*		
Trophoblast, %	49.9 (26.8–81.1)		54.6 (30.5–78.6)			0.436		
Stroma, %	9.6 (1.9–32.7)		5.5 (2.3–16.4)			0.325		
Vessels, %	3.9 (0–10.3)		3.7 (0–17.5)			0.257		

Values are shown as means and ranges. * *p* < 0.05 by Mann–Whitney a-parametric test between two groups. ^#^ *p* < 0.05 by Kruskal–Wallis a-parametric test between all 4 groups (part a and b). In C All PE are early and term cases combined, and all unaffected—are term and preterm controls-combined.

## Data Availability

The data that support the findings of this study are available from the corresponding author upon reasonable request.

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
