# Peer review of "Reduced Placental CD24 in Preterm Preeclampsia Is an Indicator for a Failure of Immune Tolerance"

_ijms, 2021, doi:10.3390/ijms22158045_

Round 1

Reviewer 1 Report

Major and minor comments

Since early- and late-onset preeclampsia have different etiology and pathomechanism, please give a comment on your Preterm preeclamptic patients (<37 weeks). Is your data show any significant difference regardind the onset of preeclampsia (<34 weeks vs. >34 weeks of gestation)?

In line 371. Please give a correct reference for updated criteria for the definition of  preeclampsia as published in June 2020

In line 90 First trimester group is later modified to First trimester and early second trimester (Line 95) group. Please clarify.

In line 89-92. 5 groups (1. first trimester, 2. third trimester of preterm, 3. normal term delivery, 4 preterm PE and 5. late PE groups mentioned which is not matching with Table 1 which shows demographic data of 4 out of the previous groups. Late PE is missing. Please clarify that there are 2 different studies were done (Austria vs. Israel)

In line 240-241. The strong statement „the level CD24 is longitudinally increased from the first trimester to term delivery as assessed by qRT-PCR analysis” is a little bit  misleading since the results are not statistically significant. See line 112. „differences indicated a tendency although none reached statistical significance”.

Major problem with the references, here some of the actual problems:

In line 86. Reference 10 is not really related to the classification of preeclamsia.

In line 253. Reference 25 and 26 is not related to the underlying pathways regarding the two types pf preeclampsia.

In line 256. Reference 27 in not by McDonald.

In line 269. Reference 28 is not by Guleria.

Author Response

Answers to Reviewer 1, Version 1

Major and minor comments

Since early- and late-onset preeclampsia have different etiology and pathomechanism, please give a comment on your Preterm preeclamptic patients (<37 weeks). Is your data show any significant difference regarding the onset of preeclampsia (<34 weeks vs. >34 weeks of gestation)?

Answer: Today, there is no universal agreement as to the differences underlying the pathways leading to the development of PE subtypes, whether term, preterm or early. The definition is mainly related to the time the clinical situation requires delivery, as the time of onset is usually not clearly defined. The pathological border line or marker differences between early cases (before 34 weeks) and preterm cases (between 34 weeks and 0 days and 36 weeks and 6 days) are very blurred. By contrast, the differences between the two subgroups of preterm and term PE are quite clear. The main differences between preterm and term PE can be found (1) with regards to blood vessel remodeling as estimating by measuring the blood flow through the uterine arteries and detected by the increased Doppler pulsatility Index (UTPI) and (2) with regards to a larger imbalance between soluble FMF-like tyrosine kinase-1 (sFlt-1) and placental growth factor (PlGF). In addition, Aspirin prevention of PE has a relative success (62%) in preterm but not term PE cases [18,28,29,30].

This comment was entered into the discussion. [Lines 300-311] We also addressed it in lines 231-235

As to our database 14 cases were early PE (delivered < 34 weeks) and 4 delivered between 34 weeks and 0 days to 36weeks and 6 days. However, there was no significant difference between the two sub-groups and none of the results pointed out to the need to sub-divide them for improving group description or clarity of the results. This is now entered into cohort description, lines 386-390 section 4.2 

In line 371. Please give a correct reference for updated criteria for the definition of preeclampsia as published in June 2020

Answer. Done (Reference 52, section 4.3 line 404-423

In line 90 First trimester group is later modified to First trimester and early second trimester (Line 95) group. Please clarify.

Answer. Corrected all the way through – first trimester was replaced to first and early second trimester (abstract and beyond lines XXX)

In line 89-92. 5 groups (1. first trimester, 2. third trimester of preterm, 3. normal term delivery, 4 preterm PE and 5. late PE groups mentioned which is not matching with Table 1 which shows demographic data of 4 out of the previous groups. Late PE is missing. Please clarify that there are 2 different studies were done (Austria vs. Israel)

Answer. Corrected. We draw clear separation between the first cohort that has 1) first and early second trimester, 2) term delivery, 3) preterm delivery, 4) preterm PE and the second cohort that has 4 third trimester groups: term delivery, term PE, preterm delivery (all with birth <34w, i.e. age-matched to early PE group) and early PE. (Abstract, results, discussion,  and methods lines 25, 30,31, 94,100, 246, 248,250, 384, 399) 

In line 240-241. The strong statement „the level CD24 is longitudinally increased from the first trimester to term delivery as assessed by qRT-PCR analysis” is a little bit  misleading since the results are not statistically significant. See line 112. „differences indicated a tendency although none reached statistical significance”.

Answer. Indeed, the sentence was modified to a less definitive statement (abstract and section 2,2 lines 27-32, 40-42, 115-122)

Major problem with the references, here some of the actual problems:

Answer. Yes,. When the journal modified the manuscript to place methods after the discussion, the entire reference numeration was completely confused. We now reordered the references and are very sorry for this.

In line 86 (now 89). Reference 10 is not really related to the classification of preeclampsia.

Answer. Right. The references are 17-18. Corrected

In line 253 (now 258). Reference 25 and 26 is not related to the underlying pathways regarding the two types pf preeclampsia.

Answer. Right. The references are 18,28,29 Corrected

In line 256 (now 261) Reference 27 in not by McDonald.

Answer. Right. McDonald reference is 30. Corrected  

In line 269. (now 271) Reference 28 is not by Guleria.

Answer. Right. McDonald reference is 32. Corrected  

Reviewer 2 Report

While reading the work, several questions arose.

  1. Did the authors compare the level of CD24 expression in the maternal and fetal parts of the placenta immunohistochemically, using PCR and Western blotting? I recommend doing this.
  2. What is known about the expression of CD 24 in macrophages, including placental macrophages?
  3. Is CD 24 expressed in the placenta in laboratory animals? Are there CD24 knockout/knockdown models?

The answers to points 2 and 3 should be provided in the discussion section.

    4. The authors present the results of the correlation analysis. However, there is no indication in the statistical analysis section what the correlation coefficient was used.

    5. The authors compared the CD 24 content in the placenta of the term control group and the placenta in preterm preeclampsia using a Western blot. It would be interesting to compare the change in the number of CD24 in early and late preeclampsia. 

Author Response

Answers to Reviewer 2, Version 1

  1. Did the authors compare the level of CD24 expression in the maternal and fetal parts of the placenta immunohistochemically, using PCR and Western blotting? I recommend doing this.

Answer: No, we did not do this in the current study. We used villous tissues (“fetal part”) of all placentas, which may contain small amounts of basal plate tissues (“maternal part”). We focused on the villous tissue as this represents the largest surface of fetal tissues (12 to 15m2 at term) in direct contact to maternal blood and the circulating immune cells. If the focus would be on extravillous trophoblast and its CD24 expression and interaction with uterine immune cells, then we would need to set up a completely different tissue collection strategy as basal plate tissue would not be sufficient for this. Then we would need placental bed biopsies, which is far from the scope of this study. This is now entered into the discussion section (lines 334-338)

  1. What is known about the expression of CD 24 in macrophages, including placental macrophages?

Answer: CD24 is expressed in human macrophages, but its role and presence on placental macrophages has not been investigated in detail. Recent studies have shown that CD24 is a potent anti-phagocytic, 'don't eat me' signal (ref. 43) that for example is capable of directly protecting cancer cells from attack by Siglec-10-expressing macrophages. Monoclonal antibody blockade of CD24–Siglec-10 signalling robustly enhances clearance of CD24+ tumours, which indicates promise for CD24 blockade in immunotherapy. Otherwise, it is known that the CD24-Siglec-10 axis can dampen immune reactions. This is the underlying rational for Israel's EXO-CD24 and Merck’s CD24-Fc as new drugs that are now tested as an imminent to control the anti COVID-19 infection related to the cytokine storm.

So far, there is no information on the expression of CD24 in placental macrophages. In our immunohistochemical analysis we did not focus on placental macrophages. However, there was nearly no specific staining in the villous stroma, indicating no or very little expression of CD24 in placental macrophages.

This is now more clearly stated in the discussion. This is now entered into the discussion section (lines 324-333)

  1. Is CD 24 expressed in the placenta in laboratory animals? Are there CD24 knockout/knockdown models?

Answer: CD24 knockout mice are available but were not reported to have birth defects. Today, only one additional study explored CD24 in the placenta of PE cases [ 27]. Our present study is thus opening a whole new line of research in evaluating the role of this protein the placenta. We hope our manuscript will stimulate other scientists to develop CD24 knockout/knockdown models to further illuminate the role of this protein in the placenta in general and in particularly, in immune tolerance during pregnancy.

This was now entered into the discussion section (lines 343-347).

The answers to points 2 and 3 should be provided in the discussion section.

Answer: The answers to points 2 and 3 have been included in the discussion section.

  1. The authors present the results of the correlation analysis. However, there is no indication in the statistical analysis section what the correlation coefficient was used.

Answer: The correlation coefficient is used in figure 2 of the results and its description. The results indicated that a respective correlation was only found in preterm PE but not in any of the other groups (lines 128-138). Description of the way the correlation coefficient was calculated was added to the statistical section (lines502-507)

  1. The authors compared the CD 24 content in the placenta of the term control group and the placenta in preterm preeclampsia using a Western blot. It would be interesting to compare the change in the number of CD24 in early and late preeclampsia.

Answer: The reviewer is absolutely right! Unfortunately, we did not have the respective tissue lysates available to run Western blots comparing early and late PE. This is why we performed immunohistochemistry comparing early and late PE with age-matched controls. This way, we could not only identify changes between the two PE groups, we were also able to identify the cells expressing CD24 in these cases.

Round 2

Reviewer 1 Report

Corrections are accepted

Reviewer 2 Report

The article could be accepted in the present form.